**Data Availability Statement:** The anonymized qualitative data is available at Figshare https://doi.org/10.6084/m9.figshare.17013881.v1.

# Experiences of COVID-19 patients admitted in a government infectious disease hospital in Nepal and its implications for health system strengthening: A qualitative study

Anup Bastola[1], Rolina Dhital [2]*, Richa Shah[2], Madhusudan Subedi[3], Pawan Kumar Hamal[4], Carmina Shrestha[2], Bimal Sharma Chalise[1], Kijan Maharjan[1], Richa Nepal[1], Sagar Rajbhandari[1]

1 Sukraraj Tropical and Infectious Disease Hospital, Kathmandu, Nepal, 2 Health Action and Research, Kathmandu, Nepal, 3 Patan Academy of Health Sciences, Lalitpur, Nepal, 4 National Academy for Medical Sciences, Kathmandu, Nepal

* rolina.dhital@gmail.com

## Abstract

### Introduction

The COVID-19 pandemic has affected the health systems in many ways. It has put unprecedented strain on health systems worldwide and exposed gaps in public health infrastructure. A health system comprises all institutions and resources working towards improving and maintaining health. Among the different aspects of health system strengthening, a patient's experiences and expectations play a crucial role in determining how well the health facilities function. This study aims to explore health system strengthening's implications based on experiences and feedback provided by COVID-19 patients admitted to a government tropical and infectious disease hospital in Nepal.

### Methods

In this qualitative study, we collected the voluntary handwritten feedback by the admitted COVID-19 patients to document the feedback and experiences from a book, maintained by the hospital. We performed thematic content analysis using the World Health Organization's six building blocks of health system as a theoretical framework which included service delivery, health workforce, information, leadership and governance, financing, and access to medicines.

### Results

Most patients in this study had positive experiences on service delivery and health workforce. Some also highlighted the gaps in infrastructure, cleanliness, and hygiene. Many suggested positive experiences on other dimensions of the health system such as financing, governance and leadership, and access to medicines reflected upon by the patients' thankfulness to the hospital and the government for the treatment they received. The responses

**Funding:** The author(s) received no specific funding for this work.

**Competing interests:** The authors have declared that no competing interests exist.

also reflected the inter-connectedness between the different building blocks of health system.

## Conclusion

This study approached a unique way to strengthen the health system by exploring patients' feedback, which suggested an overall positive impression on most building blocks of health system. However, it also highlighted certain gaps in infrastructure, cleanliness, and hygiene. It reinforces the hospital management and government's role to continue its efforts to strengthen the health system.

## Introduction

In the past year, the COVID-19 pandemic has affected the lives of people worldwide, with 253,420,051 confirmed cases and over 229,190,623 people recovered from the disease until 13 November, 2021 [1]. It has also been over one year since Nepal had its first COVID-19 patient [2]. As of 13 November, 2021, over 816,675 people had been infected by COVID-19 in Nepal, with a recovery rate above 90% [1, 3].

The COVID-19 pandemic has affected the health systems in many ways. It has led to a scarcity of human resources for health, disruption of the supply chain, increase in barriers to accessing healthcare, interference with service delivery, and spread of misinformation. They have eventually stressed the health systems and exposed gaps in public health infrastructure. It is pertinent to strengthen the health system to combat the current pandemic and prepare for future pandemics [4].

A health system comprises all institutions and resources working towards improving and maintaining health [5]. Health system strengthening involves actions taken to sustainably improve access, coverage, quality, efficiency, and accountability of the health system [6]. It also ensures that public health threats are controlled and any future outbreaks are prevented. The building blocks of health system strengthening based on the World Health Organization (WHO) are service delivery, health workforce, information, leadership and governance, financing, and access to medicines [7].

### Nepal's health system and response to COVID-19

Nepal falls into the category of low- and middle-income countries [8]. It transitioned to federalism in 2017 from a unitary government which has provided a chance to restructure the health system [9, 10]. The current COVID-19 pandemic has offered a further opportunity to adapt to the newly structured health system.

Nepal experienced its first case of COVID-19 infection on 23 January 2020 [2]. Since the beginning, the government of Nepal undertook serious actions to limit the spread of COVID-19. On March 1, 2020, a high-level coordination committee for the prevention and control of COVID-19 was formed, which was then restructured as the COVID-19 crisis management center [11]. The government also imposed a country-wide lockdown on March 24, 2020, which included a stay-at-home order for all residents and a minimum of 14 days quarantine for the infected people and those who returned from a foreign country [12, 13]. RT-PCR tests were performed on all incoming passengers at the country's only international airport, and suspected patients were transported to COVID-19 designated hospitals [14]. The government

budgeted the cost for COVID-19 response which included hospital management. The government reimbursed hospitals for free treatment provided to COVID-19 patients [15].

Sukraraj Tropical and Infectious Disease Hospital (STIDH) was the first hospital to be designated for the treatment of COVID-19 patients in Nepal [11]. It diagnosed and treated the first COVID-19 patient in the country [2]. Also, it is the only central government tropical and infectious disease hospital in Nepal. The hospital has extensive experience in treating patients with infectious diseases such as HIV, malaria, diarrheal diseases, febrile illnesses, snake bites, and tetanus [16, 17]. It is also one of the centers involved in treating infectious disease outbreaks in the past. STIDH has managed suspected cases of swine flu in 2009, dengue outbreak in different time periods, including the 2019 outbreak in Kathmandu, and cholera outbreak at different time period, including post-earthquake in 2015 [18, 19].

The hospital faced many challenges in the initial days of the pandemic. STIDH was originally a 100 bedded hospital with only two-bedded ICU. However, with the support from the government the hospital was able to upgrade its services in response to the COVID-19 pandemic in early 2020. As per the national report published in March 2021 on Nepal's response to COVID-19, the government had allocated a budget of approximately Top of Formthree million US dollars to STIDH [11]. By the time of the publication of the report, STIDH was able to utilize 88% of the allocated funds for developing infrastructure, upgrading human resources, procuring medicines and instruments, capacity development, running prevention and control programs, and other activities as needed [11].

STIDH created a 54-bedded COVID-19 ward, and added 10 beds for COVID patients in the emergency room. Simultaneously, the general ward was converted into an ICU with 23 beds [11, 20]. The number of ventilators in ICU was upgraded from two to seven [11]. Additional bilevel positive airway pressure (BiPAP) machines and high nasal flow cannulas (HNFC) were also added. A multidisciplinary team of experts in tropical and infectious diseases, internal medicine, anesthesiologists, pediatricians, orthopedic surgeons, dermatologists, and physiotherapists was formed for the management of inpatients. The hospital recruited additional human resources to sustain quality care which included appointment of three anesthesiologists for the ICU and 26 temporary nursing staff for the ICU and COVID-19 ward [11]. A counseling and a physiotherapy team were also set up to provide patient centered care for COVID-19 patients [11]. The internet connections were upgraded to improve access to information and communication [11]. All laboratory and radiological tests, and medicines were provided free of cost to COVID-19 patients [15]. The hospital also adopted a system to gather patient feedback on hospital stay before discharge with the primary goal of quality improvement.

Among the different aspects of health system strengthening for hospital management, a patient's experiences and expectations play a crucial role in determining how well the health facilities function [7, 21]. Patients' feedback and suggestions are open and lived experiences and expressions related to their hospital stay. Such input and suggestions can help improve hospital services and public health response during and after pandemics [22]. However, patients' insights and reflections are not given as much attention as their disease outcomes [23]. The experiences of the patients admitted to a central hospital could provide valuable insights for hospital administrators and policymakers [24]. Many health facilities from low resource settings can learn lessons from the experiences of STIDH during this pandemic.

Thus, this study's primary objective was to explore health system strengthening's implications based on experiences and feedback provided by COVID-19 patients admitted to a government infectious disease hospital in Nepal. The secondary objectives were to describe the overall experience of the COVID-19 patients during their stay, explore areas of improvement in hospital care, and identify lessons for future pandemics.

## Methods

### Study design

This is a qualitative study where we performed thematic content analysis using the WHO six building blocks of health system as a theoretical framework [7].

### Study setting

We collected handwritten notes from the hospital register maintained to record patient feedback at STIDH, Kathmandu, Nepal. The hospital has played a crucial role in treating a major proportion of COVID-19 patients in Nepal. As of March 2021 until the first wave, the hospital had treated 879 cases of which 408 were treated in ICU. The case fatality rate until February 2021 was 10.92.

### Data collection

The recovered COVID-19 patients had written their reflections on their experiences of hospital stay and provided feedback to the hospital at the time of discharge. The researchers (AB, SR, BSC, KM and DN) collected the anonymized pictures of the patients' handwritten notes from the hospital register. The data collection was supervised by the first author (AB) and the quality of data was ensured by other researchers not working in the hospital (RD, MS, and PKH). The included notes comprised 30 reflections of patients from the general ward and 27 from the Intensive Care Unit (ICU) and were written between January 2020 and January 2021.

The reflections were written voluntarily by the patients in Nepali and English languages. A few were also written in the Hindi language. There was no language barrier, word limit, or format for writing. All the information from the handwritten notes was transcribed to English digital text. The notes in Nepali and Hindi languages were translated to English by the research team at the time of transcribing (RD, RS, and CS). The researchers (MS and PKH) reviewed and ensured the quality of translations.

### Data analysis

The transcribed information in English was then imported to Dedoose software (version 8.3.45) for data analysis. Data analysis was done through thematic content analysis using the six building blocks of health system as a theoretical framework [7]. The categories were identified by the researchers (RD, RS, and CS) from the coding of the transcripts, which were then fitted into six major themes of building blocks of health system strengthening. The analyzed themes and categories were then shared with other authors (AB, MS, and PKH) for their review. The themes and categories were then finalized by all co-authors. Anonymous original quotes that reflected the real opinions of the respondents were chosen to give more insight.

### Ethical considerations

Ethical approval was obtained from Nepal Health Research Council (Regd. 84/2021). The permission was also obtained from STIDH to use the anonymized data of the handwritten notes of the patients from the hospital register. Informed consent from individual patients was waived as we only used anonymized data and no identifying information was included. The confidentiality of the anonymized data was strictly maintained by the research team and the information was not accessible to others except for the research team.

## Results

The findings of this study are structured around six major themes based on the WHO six core building blocks of the health system [7]. The categories for each theme included the positive reflections and feedbacks provided by the patients highlighting the areas that can be improved (Table 1).

### Service delivery

The majority of the patients reflected positive experiences on service delivery. Almost all the patients expressed their satisfaction with the service they received in terms of quality of care, person-centeredness, and basic amenities.

**Person-centeredness.** *"Everything is very good. Even nurses provided a homely environment. I hope to see and hear of similar service being provided. I am thankful to the staff here."*–ICU patient

*"It feels like I got a second life because I was timely transferred to this hospital.*

*I hope other patients in the future would continue to receive such dedication!*

*Great salute to the ICU team!"*-ICU patient.

**Quality of care.** *"Good service A++, Food A++, Service A++, Behavior A++"*–General ward patient

*"I want to thank this hospital family. All staffs, healthcare workers are providing good service. The arrangement for food and stay is good. To keep oneself safe and take care of others is a very challenging job. This has been done well and I am grateful towards the services provided."*–General ward patient.

**Basic amenities.** Some patients had a positive experience regarding basic amenities such as food provided by the hospital.

**Table 1. The six themes of health system strengthening.**

|  | Themes | Categories |
|---|---|---|
| 1. | Service Delivery | Accessibility |
|  |  | Quality |
|  |  | Person-centeredness |
|  |  | Basic amenities, cleanliness, and hygiene |
|  |  | Improving technologies |
| 2. | Health workforce | Teamwork |
|  |  | Compassion |
|  |  | Motivation |
|  |  | Communication |
| 3. | Information | Correct information |
|  |  | Misinformation |
|  |  | Access to information |
| 4. | Leadership and governance | Accountability |
|  |  | Positive impact |
|  |  | Ways to improve responses and strategies |
| 5. | Financing | Acknowledgment of positive support |
| 6. | Access to medicines | Access to timely management |

*"Food and hospital cleanliness was very good."*-General ward patient.

A few had mixed responses regarding basic amenities such as food and beverages provided by the hospital.

*"Food is ok but has to be improved in quality as I myself found hair and stones in it. Provision of tea, either milk or black, has to be done because there's no age limit of COVID patients, so choice can be given."*-ICU patient.

**Cleanliness and hygiene.** There were mixed responses regarding cleanliness and hygiene. Some patients had a positive impression about the cleanliness.

*"The cleanliness of the toilet and the ward is very good."*- General ward patient.

However, some highlighted that cleanliness and hygiene needed improvement.

*"Hygiene of cabin should be properly maintained (it's not because of staff, it's because of old rooms which need proper maintenance). Cleanliness of ICU is very good and the cleaning staff should be encouraged as much as possible."*-ICU patient.

*"Hospital's bathroom and waste disposal area is very stinky. Unhygienic bathroom may lead to secondary infections for other patients, so it has to be cleaned routinely."*-ICU patient.

**Improving technologies.** A few also suggested solutions to improve timely responsiveness by the health workers.

*"Even though with the excellent services by staffs I missed an emergency alert system which can be used by patients to call the staff whenever they required. I would suggest to have a wireless switch in the wall of each bed and its indicators with mild sound and light in the nursing station that will buzz along with the sound."*-ICU patient.

## Health workforce

**Teamwork, compassion and motivation.** The majority of the patients reflected their positive experiences towards a dedicated team of health workers and expressed their gratitude for the personalized and compassionate care.

*"After being admitted to the ICU of this hospital, I saw that all the doctors and nurses assigned to this unit for patients' treatment and care were performing their duties honestly without any pressure or fear. Doctors and nurses behaved more like a family rather than professionally towards the patients and provided treatment and care because of which all patients, including me, are very happy. The kind of treatment and service patients should receive should be just like this, with hope and trust. I heard somewhere that doctors and nurses are like god but today I could feel very happy to feel it in reality."*-ICU patient.

*"I never felt there was any god. But now I feel I was wrong, there are so many people who are blessed with god hand. Despite all the difficulties and problems, they serve people. For me they are FARISTA (angels). YES, You Guys (doctors, nurses and helpers) who are devoting their day and night during these pandemic situations are really the god hands. I feel this is my new life given by you guys. I wish I could be of any use and I can do anything from my side to assist you. I have become healthy because of you."*- ICU patient.

*"I will never forget the nurses who bathed me and shaved my beard."*- ICU patient.

Some were also sympathetic towards the difficulties faced by the health workers.

*"I am glad that I got rid of COVID-19 and since the warm services of nurses towards patients are really the best, the government should listen to their problems and obstacles as well.*

*Thank you (☺)"*–ICU patient.

**Communications.**   Many were appreciative of the health provider's communication.

*"The doctors who came on rounds were friendly and caring—exactly what every patient would want them to be. Thanks to all those working against COVID-19 directly or indirectly. We are very grateful."*- General ward patient.

However, a few highlighted some discrepancies in the communications between doctors which created confusions for the patients.

*"I found doctors' counseling to me contradicting each other. Some said you can go home after 14 days even with a positive report and some denied the statement. This literally put me in dilemma."*- ICU patient.

## Information systems

**Correct information.**   There were no reflections that directly addressed the health information system. However, some addressed the information in the social media regarding the health care for COVID-19 and compared the information they had with what they experienced.

*"I had read about this hospital just a day back on Facebook that all the health workers are good and all patients have also gone home in good health. After hearing this, I wanted to thank the hospital but as fate has it, I was meant to come myself, I came to be admitted the next day."*- General ward patient.

**Access to information.**   Some were also thankful to the hospital for providing them access to internet which kept them connected with others during the time of isolation.

*"The internet facility helped me spend my time more comfortably."*- General ward patient.

**Misinformation.**   A few also highlighted the misinformation regarding the health services provided for COVID-19 patients and shared their insights based on their own experiences.

*"The rumors against health workers not attending COVID patients at hospital were proven to be fake by all staff's (supporting, nursing staff, treating doctors) attitude, behavior and specific services provided here, 100% following the ethical part of medical service. You are great! Go on with this motive, your service is always appreciated by patients treated here."*- ICU patient.

## Governance and leadership

**Accountability.**   Many patients were appreciative of the government for their leadership and governance in providing them timely treatment.

*"I came to realize that it's not just the health facility but the government is with us too."*- General ward patient.

*"I am very satisfied with the services of Teku hospital and the department of health services."*-ICU patient.

Some also highlighted that the government must prioritize and pay more attention to this particular hospital.

*"This hospital should get number one priority from the government of Nepal to standardize the care and service delivery."*-ICU patient.

**Ways to improve responses and strategies.** A few also highlighted how government should improve their responses to the pandemic and existing strategies.

*"The probability of the infection entering the community is increasing, so there should be suggestions and pressure on the government to increase the rate of COVID testing. The inadequacy in sourcing means should be addressed timely to support the future challenges."*-ICU patient.

## Financing

**Acknowledgment of positive support.** All the patients received treatment free of cost, including free meals and medicines during their stay in the hospital. The government financed their treatment. Though none of the patients directly reflected on the free of cost treatment, many acknowledged the service provided by the government and were thankful for the service they received.

*"I am proud of Nepal Government that they looked after us for 15 days."*- General ward patient.

*"I have not a single sentence to comment or suggest because the service this government hospital is giving to the patients is simply the best."*-ICU patient.

*"When we were admitted to the ward, what we had in mind was that we would have to adjust to the government provided services- whatever quality those would be. But it turned out different. We are actually very pleased with the services that were provided here."*- General ward patient.

## Access to medicines

**Timely management.** As a central government infectious disease hospital, all the patients had access to essential medicines. However, only a few reflected upon their treatment approaches.

*"I was admitted to the hospital due to severe pneumonia. But I was treated successfully with plasma therapy and all other technologies. I am thankful to all the health workers for giving me a new life and for all the love."*–General ward patient.

## Discussion

In this study, the experiences of the patients admitted to a government infectious disease hospital in Nepal provide an insight into the different dimensions of the building blocks of the health system [7]. The majority of the patients in this study had positive experiences on service delivery and health resources reflected by the hospital teams' competent and compassionate care. However, the study also highlights the gaps in infrastructure, cleanliness, and hygiene, which are also important elements of service delivery and the health system in general [6]. The study also suggested positive experiences on other dimensions of the health system such as financing, governance, and leadership. The responses also reflected the interconnectedness between the different building blocks of the health system.

The high level of satisfaction among the patients about the service delivery in this study reflects the hospital's efforts and preparation. Service delivery encompasses quality care, accessibility, and person-centeredness [24]. It also includes the provision of basic amenities, cleanliness, and hygiene [6]. Most LMICs have suffered from poor service delivery during this pandemic attributed to the shortage in human resources, medicines, diagnostics, and other technologies [25]. STIDH, as a government hospital, had been facing similar challenges related to low resources for many years before the pandemic. However, under good leadership, the hospital could efficiently upgrade the service delivery amid the pandemic and resource crisis in a short span of time. The patients' experiences in the study reflected person-centered care provided in the hospital through good communication along with attending to patients' personal needs beyond regular treatment. Person-centered care enables shared decision-making through stronger provider-patient relationships and effective communications. Such shared decision making and person-centered approach could play a crucial role in reducing health inequities which may eventually improve the service delivery for COVID-19 patients [26].

However, the challenges related to infrastructures remained visible despite the efforts as highlighted by patients' mixed responses. Poor infrastructure has remained a major challenge even before the pandemic in most LMICs, including Nepal [27]. Before the pandemic, the hospital in this study was only equipped with basic facilities with just two ICU beds and no isolation wards for a disease outbreak of such a massive scale [11]. Even though the hospital could upgrade most of the services, more support and efforts could be needed to further strengthen the infrastructure to improve ongoing services and prepare for future pandemics. The patients also reflected mixed responses on basic amenities such as food and beverages. The government had allocated funds to provide food and beverages for the COVID-19 patients in the designated hospitals [28]. The hospital served food four times a day, including breakfast with vegetarian and non-vegetarian options, and food customized based on patients' health status. Such provision of free meals for the admitted patients in a government hospital is noteworthy. Nevertheless, the feedback from the patients could provide an opportunity for the hospital to improve the shortcomings.

Some patients in this study also had complaints regarding cleanliness and hygiene, particularly of the hospital toilets. Patients' satisfaction with the service delivery is largely influenced by their impression of the cleanliness of the hospital [29, 30]. They play an important role in sensitizing hospital administrators on the shortcomings of service delivery [31]. However, it is quite common for most hospitals to minimize the maintenance costs related to hygiene,

cleaning products, and the training of their human resources [31]. When the cleanliness and hygiene are compromised, the subsequent cost of its consequences could be much higher as it may lead to many hospital-acquired infections including the spread of COVID-19. Coronavirus has been detected in stool samples; hence during flushing, the fecal matter may become aerosolized and can be inhaled [32–34]. This aerosol can settle on surfaces which can cause transmission of the virus. This can be prevented by frequent cleaning and disinfection of surfaces, increasing ventilation, closing the lid when flushing the toilet, and hand washing [34, 35]. The patients' feedback on cleanliness and hygiene also emphasizes training the cleaning staff on cleanliness, hygiene, and efficient disinfection procedures during the pandemic [36].

In this study, all the patients were happy with the services provided by the health workforce of the hospital and acknowledged the good teamwork. The health workforce comprises a diverse team of professionals who are integral to a health system's functioning. It includes the clinical staff such as doctors, nurses, pharmacists, laboratory scientists, and health technicians, and management and support staff who may not directly deliver the services [5]. Despite the scarcity of human resources, the hospital doctors conducted a minimum of three rounds per day in-person and continuous monitoring remotely, and nursing care was available all the time. Throughout Nepal, the healthcare staff have worked selflessly, making the best use of available resources [37]. The healthcare providers faced a double burden of increased workload because of scarcity of health staff and exposure to COVID-19 infection [38]. The numbers of health workforce have a direct and positive association with people's health outcomes [39, 40]. Even STIDH, a central hospital, had to recruit temporary staff for the pandemic's peak duration to cater to patients' increased health needs [11]. Nevertheless, the recruitment of temporary staff could be considered a smart strategy to improve the hospital's health system as it helped deliver satisfactory services to the COVID-19 patients. This also brings light to the need to acknowledge and motivate the human resources for the work they have been doing. Also, the healthcare staff should be trained and prepared for future pandemics with mock drills for emergency response [41].

The health information system also plays a crucial role in health system through its key functions on data generation, compilation, analysis and synthesis, and communications [7]. While it was not possible to obtain much information on data generation through patients' perspectives, their reflections included positive experiences on access to communication. Most patients in the study had access to correct information, but some addressed the circulating misinformation in social media about healthcare staff not attending patients. Patients were grateful to have an internet connection during their stay at the hospital, which helped them stay connected with their families and be updated with the news. A systematic approach to communication is identified to improve the experience of COVID-19 patients significantly [21]. Receiving regular updates from nurses and physicians several times a day is one of the important factors. Allowing internet connection and encouraging video chats can be another way of improving patients' psychosocial issues. Reassurance to patients that the restriction is part of the job is another factor that needs to be addressed. Encouraging patients not to hesitate to ask for help is another important domain that can help improve service delivery [21]. Communication with caretakers and relatives with improved access to internet could have been key factors that were addressed at STIDH that led to more patients' satisfaction.

As STIDH was the first designated COVID-19 hospital and a central government hospital, patients' experiences from the hospital reflect the accountability and actual implementation of the health financing, leadership, and governance in Nepal. Health financing is fundamental to the performance and sustainability of health systems. It comprises a complex system of collecting and allocating funds to ensure that all individuals can access effective public health and personal health care. Health financing is also directly linked to leadership and governance [7].

Leadership and governance involve ensuring the proper implementation of strategic policy frameworks, designing systems, coalition-building, and accountability [7]. The government had allocated a budget for the government health facilities across Nepal to provide treatment to COVID-19 patients free of cost [15]. Although elaborate information was not available, most patients were thankful to the hospital and government's leadership for their treatment. Patients admitted to COVID-19 designated hospitals did not face financial catastrophe because of out-of-pocket payments. The finding is a good indicator that financing, leadership, and governance in response to COVID-19 had been positive. However, because of a lack of hospital and ICU beds in these hospitals, many patients had to seek care in private hospitals, which led to high expenditure and, for some, inability to receive hospital care because of financial constraints [42–44]. This explains the importance of the government's role in pandemics and the need to focus on more equitable financing and preparedness for future emergencies.

The government of Nepal has been providing the essential medicines to government and public health facilities across the country even before the pandemic [45]. During the pandemic, the patients had timely access to life saving medicines for COVID-19 complications, as reflected in this study. A well-functioning health system requires equitable access to essential medical products, vaccines, and technologies. Moreover, the efficacy, cost-effectiveness, safety, and quality must be ensured and should be evidence-based [11]. Though limited information is available on patient's individual experiences, the positive responses on timely access to medicines and treatment is encouraging. Government should continue to assure that all essential medicines are stockpiled and easily accessible for free by those in need.

This study has some limitations. As a qualitative study from a single setting, the findings only reflect a smaller population's perspective, which may not be generalized to the entire population. As the patients were not directly interviewed, the reflections may not truly represent their complete experience of hospital stay. Also, the patients were aware that their reflections would be read by others and they were not told to anonymize their names in the feedback register, which could have led to social desirability bias. Moreover, the findings don't reflect the experiences of the patients who did not recover. The perspectives of the deceased patients' families could also have been insightful which was not included in this study. Furthermore, this study only reflects the experiences of the patients and the hospital from the first wave in Nepal until March, 2021. The health facilities faced severe challenges in managing the patients during the second wave between April and June 2021 [46]. As this study was already completed before the second wave, it was beyond the scope of this study to reflect the experiences of second wave. The experiences and challenges of the second wave in Nepal warrants a separate study to look deeper into different dimensions of managing the different peaks of a pandemic.

Despite the limitations, this study provides fresh insights as the patients reflected upon their experience during hospital stay and treatment right before their discharge. Thus, this study is not prone to recall bias. Few studies on COVID-19 have explored patients' perspectives and linked it to health system strengthening.

It is evident from this study that the leadership and management of the hospital had played an important role for service delivery in the first wave of the pandemic. The findings also reflect the importance of focusing on all the six building blocks of health system strengthening to provide services to the patients [7]. In particular, the availability of funding from the government, efficient implementation of plans, proper infrastructure, and motivated health workforce had been crucial for this central hospital from a resource constraint setting to continue provide quality services amid the uncertainties of the pandemic.

## Conclusion

This study approached a unique way to strengthen the health system by exploring patients' feedback from their experience of hospital stay during treatment for COVID-19. It reinforces the role of the government to provide timely and necessary health services to the population in general and specifically during the pandemic.

## Author Contributions

**Conceptualization:** Anup Bastola, Rolina Dhital, Madhusudan Subedi.

**Data curation:** Anup Bastola, Bimal Sharma Chalise, Kijan Maharjan, Richa Nepal, Sagar Rajbhandari.

**Formal analysis:** Rolina Dhital, Richa Shah, Madhusudan Subedi, Pawan Kumar Hamal, Carmina Shrestha.

**Investigation:** Anup Bastola, Rolina Dhital, Richa Shah, Madhusudan Subedi, Pawan Kumar Hamal, Carmina Shrestha, Bimal Sharma Chalise, Kijan Maharjan, Richa Nepal, Sagar Rajbhandari.

**Methodology:** Anup Bastola, Rolina Dhital, Richa Shah, Madhusudan Subedi, Pawan Kumar Hamal, Carmina Shrestha, Kijan Maharjan, Richa Nepal, Sagar Rajbhandari.

**Project administration:** Anup Bastola, Bimal Sharma Chalise, Kijan Maharjan, Richa Nepal, Sagar Rajbhandari.

**Resources:** Anup Bastola, Richa Shah, Madhusudan Subedi, Pawan Kumar Hamal, Carmina Shrestha, Bimal Sharma Chalise, Kijan Maharjan, Richa Nepal, Sagar Rajbhandari.

**Software:** Rolina Dhital, Richa Shah, Pawan Kumar Hamal, Carmina Shrestha.

**Supervision:** Anup Bastola, Madhusudan Subedi, Pawan Kumar Hamal, Sagar Rajbhandari.

**Validation:** Anup Bastola, Madhusudan Subedi, Pawan Kumar Hamal, Bimal Sharma Chalise, Kijan Maharjan, Sagar Rajbhandari.

**Visualization:** Anup Bastola, Rolina Dhital, Madhusudan Subedi, Bimal Sharma Chalise.

**Writing – original draft:** Anup Bastola, Rolina Dhital, Richa Shah.

**Writing – review & editing:** Anup Bastola, Rolina Dhital, Richa Shah, Madhusudan Subedi, Pawan Kumar Hamal, Carmina Shrestha, Bimal Sharma Chalise, Kijan Maharjan, Richa Nepal, Sagar Rajbhandari.

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
