## [Decision Letter · Decision Letter 0]

11 Nov 2021

PONE-D-21-11970Experiences of COVID-19 patients admitted in a government infectious disease hospital in Nepal and its implications for health system strengthening: A qualitative studyPLOS ONE

Dear Dr. Dhital,

Thank you for submitting your manuscript to PLOS ONE. After careful consideration, we feel that it has merit but does not fully meet PLOS ONE’s publication criteria as it currently stands. Therefore, we invite you to submit a revised version of the manuscript that addresses the points raised during the review process.

Kindly revise your manuscript as per the reviewer's comments. Also ensure the requirements for data avaialbility of the journal are met. 

We look forward to receiving your revised manuscript.

Kind regards,

Pathiyil Ravi Shankar

Academic Editor

PLOS ONE

Journal Requirements:

Reviewers' comments:

Reviewer's Responses to Questions

**Comments to the Author**

1. Is the manuscript technically sound, and do the data support the conclusions?

Reviewer #1: Yes

Reviewer #2: Yes

Reviewer #3: Yes

2. Has the statistical analysis been performed appropriately and rigorously? 

Reviewer #1: N/A

Reviewer #2: N/A

Reviewer #3: No

3. Have the authors made all data underlying the findings in their manuscript fully available?

Reviewer #1: Yes

Reviewer #2: No

Reviewer #3: No

4. Is the manuscript presented in an intelligible fashion and written in standard English?

Reviewer #1: Yes

Reviewer #2: Yes

Reviewer #3: Yes

5. Review Comments to the Author

Reviewer #1: Since this is qualitative study, it focuses on the concepts which is acquired through the patients feedback so there is no statistical analysis for interpretation. Basically it describes the function of a hospital in Covid-19 crisis on the basis of WHO six core building blocks of the health system. however, it is not the unique study but it helps to strengthen the health system and

as stated in the conclusion. The number of participants is small but their subjective views may not relate to reality. There is question of bias to exclude. It is well written in English with open-ended patients feedback. Ethical approval was obtained from the national authority of research. it meets the criteria for publication.

Reviewer #2: The reviewer thanks the authors for this important work.

The reviewer is aware of the exemplary service during COVID-19 waves that was provided by the center in which this study has been carried out; hence, agrees that the experiences from the center merit sharing so that other centers in similar situation in LMICs are informed.

The reviewer has some suggestion on the manuscript:

In the abstract and the main body of manuscript, the use of the phrase 'handwritten notes' has not always clearly depicted the true nature of source document, which is a voluntary non-mandatory feedback book to be handwritten by the patients in order to document the feedback, maintained by the hospital. Some language and vocabulary change may be required wherever this is the case.

The very nature of the source documents entails an inherent possibility that the patients who were satisfied with the services are more likely to provided the handwritten feedback whereas the proportion for the dissatisfied ones could be less. The reviewer assumes that the patients were not told to anonymize themselves while providing the data, this could also inhibit the patients from providing a feedback of dissatisfaction. The reviewer, however, also acknowledges the fact that there are responses of dissatisfaction documented. Hence the reviewer advises to add this aspect as one of the limitations.

There is perhaps one more limitation: the experiences of those who did not recover is probably not covered by the study. Feedback responses from the family member could have supplemented this. This deficiency could also be mentioned.

One useful information to add is the total number of admissions in ICU and General ward, and the number of recovered patients (if available) during the study duration.

After completion of the study date (April) there had been a second wave of COVID 19 in the country that had been even more devastating. It may be worthwhile to share some of the experiences from second wave, if relevant to this manuscript, in the discussion section.

As evident from the manuscript, the availability of (funding from government), efficient implementation of plans including infrastructure, motivated workforce have been crucial in the delivery of services from this center. Perhaps the managerial and clinical leadership was effective in achieving this. This aspect could be highlighted (to the extent they can be supported by facts) in discussion or conclusion, as these could be the determinants for a successful care delivery model for other places as well.

The statement in line 117-118 merits a citation.

The basic definitions, such as the ones in line 213-14, could be kept to minimum to keep the manuscript concise.

The reviewer is not very sure about the requirement for data sharing for qualitative studies, hence requests the editor to examine the response on "Have the authors made all data underlying the findings in their manuscript fully available?" (number 3)

Overall, the reviewer is of the opinion that the manuscript merits to be published.

Thank you.

Reviewer #3: Dear Authors

Article on COVID 19 Experiences of COVID-19 patients admitted in a government infectious disease hospital in Nepal and its implications for health system strengthening: A qualitative study makes interesting reading

As it qualitative review by Patient response handwritten notes It has some bias which has been sighted as limitations

I have some queries.

whether patients in ICU & critically sick were included or not ? what point time of time response were recorded ? at discharge or follow up,

Since it is not direct interview whether any supervision was done ?

whether any scales were used for recording response whether data analysis was done?

similar study was published form Nepal

Bhatt, N., Bhatt, B., Gurung, S., Dahal, S., Jaishi, A. R., Neupane, B., & Budhathoki, S. S. (2020). Perceptions and experiences of the public regarding the COVID-19 pandemic in Nepal: a qualitative study using phenomenological analysis. BMJ open, 10(12), e043312. https://doi.org/10.1136/bmjopen-2020-043312 how different is this study

6. PLOS authors have the option to publish the peer review history of their article (what does this mean?). If published, this will include your full peer review and any attached files.

Reviewer #1: **Yes: **Dr. Rupesh Mukhia

Reviewer #2: **Yes: **Bishnu Rath Giri

Reviewer #3: **Yes: **Dr. M.Mukhyaprana Prabhu

---

## [Author Response · Author response to Decision Letter 0]

15 Nov 2021

Dear Editor and Reviewers,

Thank you for your encouraging comments and the opportunity to revise our manuscript.

Below please find our responses to each comment.

Editor's comments

Authors' response

We have formatted the manuscript as per the PLOS ONE's style requirements, including those for file naming.

Authors' response

We have reviewed the references list, removed a reference of which the web link was no longer available, and updated the reference list.

Authors' response

We have uploaded the anonymized qualitative data to Figshare https://doi.org/10.6084/m9.figshare.17013881.v1

Reviewers' comments:

Reviewer #1: Since this is qualitative study, it focuses on the concepts which is acquired through the patients feedback so there is no statistical analysis for interpretation. Basically it describes the function of a hospital in Covid-19 crisis on the basis of WHO six core building blocks of the health system. however, it is not the unique study but it helps to strengthen the health system and

as stated in the conclusion. The number of participants is small but their subjective views may not relate to reality. There is question of bias to exclude. It is well written in English with open-ended patients feedback. Ethical approval was obtained from the national authority of research. it meets the criteria for publication.

Authors' response

Thank you for your encouraging comments.

Reviewer #2: The reviewer thanks the authors for this important work.

The reviewer is aware of the exemplary service during COVID-19 waves that was provided by the center in which this study has been carried out; hence, agrees that the experiences from the center merit sharing so that other centers in similar situation in LMICs are informed.

Authors' response

Thanks for your positive comment.

The reviewer has some suggestion on the manuscript:

In the abstract and the main body of manuscript, the use of the phrase 'handwritten notes' has not always clearly depicted the true nature of source document, which is a voluntary non-mandatory feedback book to be handwritten by the patients in order to document the feedback, maintained by the hospital. Some language and vocabulary change may be required wherever this is the case.

Authors' response

Thanks for the suggestion. We have revised it as suggested to provide more clarity in Abstract, Lines 26-28.

The very nature of the source documents entails an inherent possibility that the patients who were satisfied with the services are more likely to provided the handwritten feedback whereas the proportion for the dissatisfied ones could be less. The reviewer assumes that the patients were not told to anonymize themselves while providing the data, this could also inhibit the patients from providing a feedback of dissatisfaction. The reviewer, however, also acknowledges the fact that there are responses of dissatisfaction documented. Hence the reviewer advises to add this aspect as one of the limitations.

Authors' response

Thanks for the advice. We have mentioned the possibility of social desirability in the manuscript. We have further clarified that their names were not anonymized in the feedback register in the revised manuscript. ( Discussion, Lines 421-423)

 “Also, the patients were aware that their reflections would be read by others and they were not told to anonymize their names in the feedback register, which could have led to social desirability bias.”

There is perhaps one more limitation: the experiences of those who did not recover is probably not covered by the study. Feedback responses from the family member could have supplemented this. This deficiency could also be mentioned.

Authors' response

We have now added this point as a limitation in the revised manuscript. (Discussion, Lines 423-425)

“Moreover, the findings don’t reflect the experiences of the patients who did not recover. The perspectives of the deceased patients’ families could also have been insightful which was not included in this study.”

One useful information to add is the total number of admissions in ICU and General ward, and the number of recovered patients (if available) during the study duration.

Authors' response

We have added this information in the Methods section under study settings in the revised manuscript. (Methods, Lines134-136)

After completion of the study date (April) there had been a second wave of COVID 19 in the country that had been even more devastating. It may be worthwhile to share some of the experiences from second wave, if relevant to this manuscript, in the discussion section.

Authors' response

This study was conducted and submitted to the journal in March 2021, before the second wave. Therefore, this study doesn’t cover the experiences of second wave. We have added this point in the limitation that this study only reflects the experiences from the first wave. The experiences for the hospital and the patients in the second wave was far more challenging which warrants a separate study. (Discussion, Lines 425-431)

As evident from the manuscript, the availability of (funding from government), efficient implementation of plans including infrastructure, motivated workforce have been crucial in the delivery of services from this center. Perhaps the managerial and clinical leadership was effective in achieving this. This aspect could be highlighted (to the extent they can be supported by facts) in discussion or conclusion, as these could be the determinants for a successful care delivery model for other places as well.

Authors' response

We have added a paragraph in the discussion highlighting these points it in the revised manuscript as suggested.( Lines 436-442)

The statement in line 117-118 merits a citation.

Authors' response

We have added a citation as suggested.

The basic definitions, such as the ones in line 213-14, could be kept to minimum to keep the manuscript concise.

Authors' response

We have now removed the basic definitions for each theme (Lines 171-172, 213-14, 247-248, 269-270, 290-291, and 34-305) from the original manuscript to keep the manuscript concise as suggested.

The reviewer is not very sure about the requirement for data sharing for qualitative studies, hence requests the editor to examine the response on "Have the authors made all data underlying the findings in their manuscript fully available?" (number 3)

Authors' response

We have uploaded anonymized qualitative data to Figshare.

Overall, the reviewer is of the opinion that the manuscript merits to be published.

Authors' response

Thank you for your encouraging comment!

Reviewer #3: Dear Authors

Article on COVID 19 Experiences of COVID-19 patients admitted in a government infectious disease hospital in Nepal and its implications for health system strengthening: A qualitative study makes interesting reading

As it qualitative review by Patient response handwritten notes It has some bias which has been sighted as limitations

I have some queries.

whether patients in ICU & critically sick were included or not ? what point time of time response were recorded ? at discharge or follow up,

The patients from ICU and critically sick were also included in this study. 

Authors' response

The data was collected at the time of discharge and the patients from ICU were also included in this study. We have mentioned in the Methods under Data collection (Lines 139, 141-143)

Since it is not direct interview whether any supervision was done?

Authors' response

We have added this information in the methods section under data collection (Lines 141-142)

“The data collection was supervised by the first author (AB) and the quality of data was ensured by other researchers not working in the hospital (RD, MS, and PKH).”

Whether any scales were used for recording response whether data analysis was done?

Authors' response

We performed the thematic content analysis based on the six building blocks of WHO. Therefore, scales were not used.

similar study was published form Nepal

Bhatt, N., Bhatt, B., Gurung, S., Dahal, S., Jaishi, A. R., Neupane, B., & Budhathoki, S. S. (2020). Perceptions and experiences of the public regarding the COVID-19 pandemic in Nepal: a qualitative study using phenomenological analysis. BMJ open, 10(12), e043312. https://doi.org/10.1136/bmjopen-2020-043312 how different is this study

Authors' response

Thanks for sharing this insightful article. However, our study is different from this study in terms of objectives and research participants. This study has explored the perceptions of public on their general understanding of COVID-19, disease prevention, source of information and misconceptions, expectation and challenges; and personal and societal consequences of COVID-19, social distancing and lockdown.

Our study was focused on the experiences of COVID-19 patients who were admitted in the hospital and highlights their experience of care. The findings are structured around the six building blocks of health system. Our study is more about the strengthening of health system with a focus on health facilities by exploring patients’ feedback on health services delivered to them. The two studies both focused on COVID-19 but on different dimensions and complement each other.

---

## [Decision Letter · Decision Letter 1]

6 Dec 2021

Experiences of COVID-19 patients admitted in a government infectious disease hospital in Nepal and its implications for health system strengthening: A qualitative study

PONE-D-21-11970R1

Dear Dr. Dhital,

We’re pleased to inform you that your manuscript has been judged scientifically suitable for publication and will be formally accepted for publication once it meets all outstanding technical requirements.

Kind regards,

Pathiyil Ravi Shankar

Academic Editor

PLOS ONE

Additional Editor Comments (optional):

Reviewers' comments:

Reviewer's Responses to Questions

**Comments to the Author**

1. If the authors have adequately addressed your comments raised in a previous round of review and you feel that this manuscript is now acceptable for publication, you may indicate that here to bypass the “Comments to the Author” section, enter your conflict of interest statement in the “Confidential to Editor” section, and submit your "Accept" recommendation.

Reviewer #3: All comments have been addressed

Reviewer #4: (No Response)

2. Is the manuscript technically sound, and do the data support the conclusions?

Reviewer #3: Yes

Reviewer #4: Yes

3. Has the statistical analysis been performed appropriately and rigorously? 

Reviewer #3: N/A

Reviewer #4: Yes

4. Have the authors made all data underlying the findings in their manuscript fully available?

Reviewer #3: Yes

Reviewer #4: Yes

5. Is the manuscript presented in an intelligible fashion and written in standard English?

Reviewer #3: Yes

Reviewer #4: Yes

6. Review Comments to the Author

Reviewer #3: Good qualitative study focusing on patient perspective on health care delivery during COVID pandemic in resource poor settings.

Some observations:

mortality of 10 % was high any detailed analysis was done

Reviewer #4: Authors have mentioned the required info needed for the qualitative study and written manuscript following guidelines of qualitative research.

7. PLOS authors have the option to publish the peer review history of their article (what does this mean?). If published, this will include your full peer review and any attached files.

Reviewer #3: **Yes: **Mukhyaprana Manuru Prabhu

Reviewer #4: No

---

## [Editor Report · Acceptance letter]

13 Dec 2021

PONE-D-21-11970R1 

Experiences of COVID-19 patients admitted in a government infectious disease hospital in Nepal and its implications for health system strengthening: A qualitative study 

Dear Dr. Dhital:

I'm pleased to inform you that your manuscript has been deemed suitable for publication in PLOS ONE. Congratulations! Your manuscript is now with our production department. 

Kind regards, 

on behalf of

Dr. Pathiyil Ravi Shankar 

Academic Editor

PLOS ONE